# AUNIP/C1orf135 directs DNA double-strand breaks towards the homologous recombination repair pathway

Jiangman Lou[1], Hongxia Chen[1,2], Jinhua Han[1], Hanqing He[1], Michael S.Y. Huen[3], Xin-hua Feng[1], Ting Liu[4] & Jun Huang [1]

DNA double-strand breaks (DSBs) are mainly repaired by either homologous recombination (HR) or non-homologous end-joining (NHEJ). Here, we identify AUNIP/C1orf135, a largely uncharacterized protein, as a key determinant of DSB repair pathway choice. AUNIP physically interacts with CtIP and is required for efficient CtIP accumulation at DSBs. AUNIP possesses intrinsic DNA-binding ability with a strong preference for DNA substrates that mimic structures generated at stalled replication forks. This ability to bind DNA is necessary for the recruitment of AUNIP and its binding partner CtIP to DSBs, which in turn drives CtIP-dependent DNA-end resection and HR repair. Accordingly, loss of AUNIP or ablation of its ability to bind to DNA results in cell hypersensitivity toward a variety of DSB-inducing agents, particularly those that induce replication-associated DSBs. Our findings provide new insights into the molecular mechanism by which DSBs are recognized and channeled to the HR repair pathway.

[1] Life Sciences Institute and Innovation Center for Cell Signaling Network, Zhejiang University, Hangzhou, Zhejiang 310058, China. [2] Huadong Research Institute for Medicine and Biotechniques, Nanjing, Jiangsu 210002, China. [3] School of Biomedical Sciences, Li Ka Shing Faculty of Medicine, The University of Hong Kong, Hong Kong Special Administrative Region, Pok Fu Lam, Hong Kong, China. [4] Department of Cell Biology, Zhejiang University School of Medicine, Hangzhou, Zhejiang 310058, China. Jiangman Lou and Hongxia Chen contributed equally to this work. Correspondence and requests for materials should be addressed to J.H. (email: jhuang@zju.edu.cn)

DNA double-strand breaks (DSBs) are the most deleterious form of DNA damage, which if unrepaired or repaired incorrectly, can contribute to various genetic disorders including cancer, neurodegeneration, and immunodeficiency[1]. DSBs can arise as a result of errors during DNA replication, and can be induced by exogenous DNA-damaging agents including ionizing radiation (IR) and various chemotherapeutic drugs[1]. DSBs are mainly repaired via two pathways–non-homologous end joining (NHEJ) and homologous recombination (HR), both of which are highly conserved from yeast to human[2–6]. NHEJ is a relatively fast and simple process that involves direct end-to-end ligation of the DSB ends, and this pathway is active throughout interphase[7–9]. The key players in NHEJ include the DNA end-binding Ku70/80 heterodimer, the DNA-dependent protein kinase catalytic subunit (DNA-PKcs), X-ray cross-complementing protein 4 (XRCC4), XRCC4-like factor (XLF), DNA ligase IV, and the newly identified PAXX (a paralog of XRCC4 and XLF)[7, 8, 10, 11]. In contrast to NHEJ, HR is a complex, multi-step repair pathway that requires the sequential activity of a cohort of proteins and occurs primarily in the S and G2 phases of the cell cycle[2–6]. HR relies on the presence of a sister chromatid as a donor template, and is initiated by nuclease-mediated extensive 5′-3′ resection of DSB ends, resulting in long stretches of 3′ single-stranded DNA (ssDNA) that subsequently invades the homologous duplex DNA[12–14]. It is now well-established that DSBs are resected in a two-step manner[12–14]. Initially, the evolutionarily-conserved MRE11-RAD50-NBS1/XRS2 (MRN/X) complex and its associated factor CtIP/Sae2 carry out limited resection near the break site to generate a short 3′ overhang[15–19]. The partially-resected DNA is further processed by two parallel pathways; one that is dependent on the 5′-3′ exonuclease Exo1 and the other dependent on the concerted action of the BLM/Sgs1 helicase and the Dna2 endonuclease[20–23].

While both NHEJ and HR machineries can repair DSBs, choice of the more appropriate DSB repair pathway is key to maintenance of genome stability, especially at the organismal level[24, 25]. To date, a number of determinants have been reported to influence the choice between the two pathways. One of these is cell cycle[24, 26–28]. Studies have shown that efficient DNA end processing is restricted to the S and G2 phases, and is regulated by cell cycle-dependent CDK activity[24, 26–30]. By preventing HR outside of the S and G2 phases of the cell cycle, exchanges between homologous chromosomes are reduced, cells thereby suppress DSB-associated loss of heterozygosity and chromosomal rearrangements[24, 26, 27]. In addition to effects arising from cell cycle, the nature of DSBs also influences choice of repair pathways[24, 31–33]. Indeed, although both NHEJ and HR contribute to repair of X- or γ-ray-induced two-ended DSBs in the S

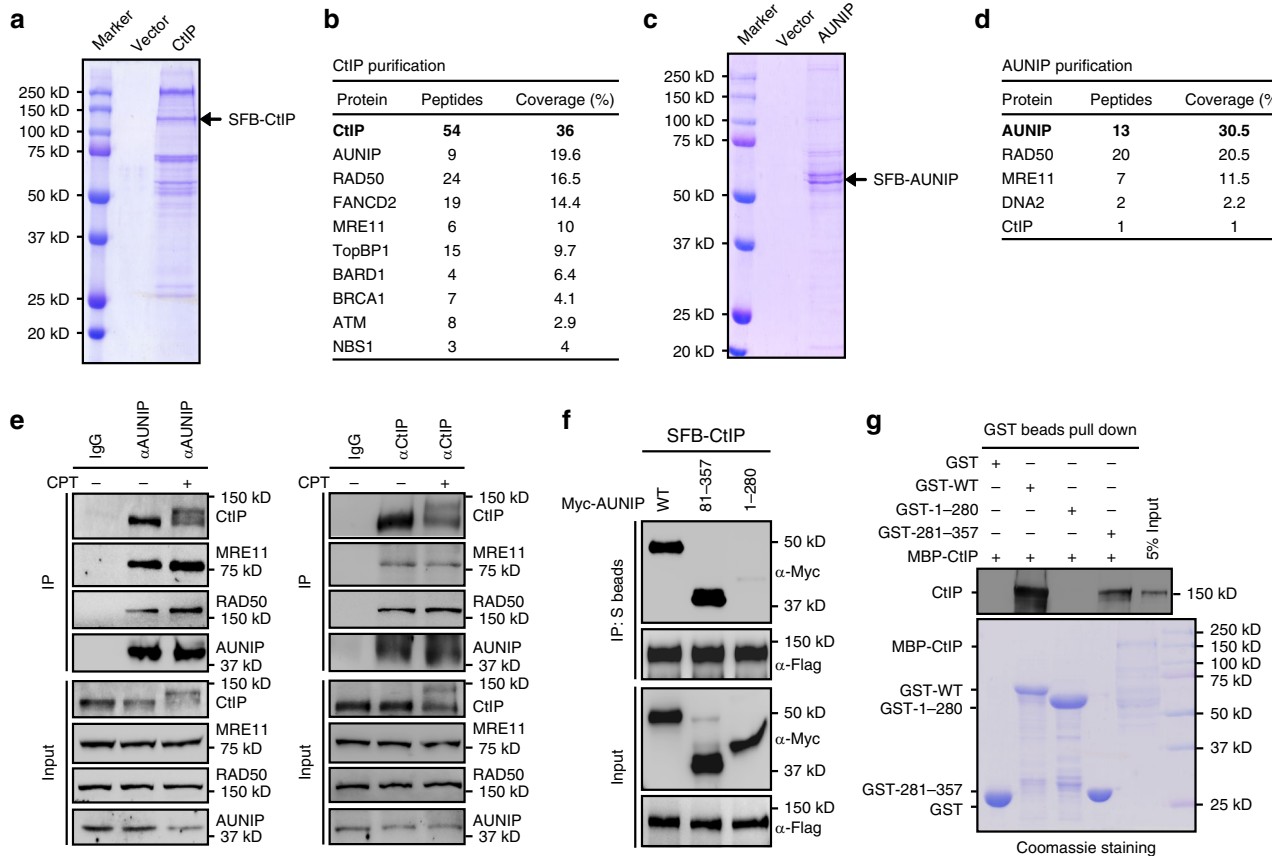

**Fig. 1** Identification of AUNIP as a CtIP-associated protein. **a**, **c** CtIP (**a**) or AUNIP (**c**) protein complexes separated by SDS-PAGE were stained with Coomassie blue. **b**, **d** Proteins identified by mass spectrometric analyses of CtIP or AUNIP protein complexes are listed. Bait proteins are indicated in bold letters. **e** AUNIP interacts with CtIP and the MRN complex. HeLa cell lysates treated with Benzonase were precipitated with anti-AUNIP or anti-CtIP antibodies and were analyzed by immunoblotting with indicated antibodies. **f** AUNIP interacts with CtIP via its C-terminus. HEK293T cells transfected with indicated plasmids were lysed with NETN buffer 24 h post transfection. Cell lysates were then incubated with S-protein beads and immunoprecipitated proteins were analyzed by immunoblotting experiments using indicated antibodies. **g** Direct in vitro binding between recombinant GST-AUNIP and MBP-CtIP purified from *E. coli*. GST served as a negative control for CtIP binding. Top: CtIP was detected by immunoblotting experiment. Bottom: purified proteins were visualized by Coomassie staining

and G2 phases in mammalian cells[24, 31–33], HR-deficient cells are much more tolerant to irradiation, indicating that NHEJ likely plays a more important role in the repair of two-ended DSBs[33–36]. By contrast, replication-associated one-ended DSBs are repaired almost exclusively by HR[37, 38]. In support of this working model, NHEJ is accountable for the genome instability and cell cytotoxicity phenotypes in HR-deficient cells when challenged with agents known to induce replication-associated DSBs, including camptothecin (CPT, a DNA topoisomerase I inhibitor) and poly(ADP-ribose) polymerase (PARP) inhibitors[39–41]. Nevertheless, it remains unclear how the nature of DSB determines usage of DSB repair pathways.

In this study, we used an affinity purification approach to isolate CtIP-containing protein complexes, and have identified AUNIP/C1orf135 as a primary determinant of DSB repair pathway choice. We show that AUNIP is recruited to DNA damage sites through a DNA-binding motif that shows a strong binding preference for DNA substrates that mimick structures generated at stalled replication forks. We further demonstrate that AUNIP physically interacts with CtIP and is required for efficient CtIP concentration at DNA lesions. Consequently, loss of AUNIP, or ablation of its ability to bind to DNA or CtIP, impaired CtIP-dependent DNA-end resection, compromised HR repair, and resulted in cell hypersensitivity toward a variety of DSB-inducing agents, particularly those that induce replication-associated DSBs. Our results support a model in which AUNIP serves as a sensor of DNA damage, anchoring CtIP to DSB sites to drive CtIP-dependent DNA end resection and ensuing HR repair.

## Results

**Identification of AUNIP as a CtIP-associated protein**. To further elucidate the role of CtIP in DNA end resection and DSB repair pathway choice, we performed tandem affinity purification (TAP) using a HEK293T cell line that stably expresses SFB-tagged (S-protein tag, Flag epitope tag, and streptavidin-binding peptide tag) wild-type CtIP to isolate proteins that associate with CtIP (Fig. 1a). Mass spectrometric analysis identified a number of previously reported CtIP-interacting proteins, including the MRN (MRE11-RAD50-NBS1) complex, FANCD2, TopBP1, ATM, BRCA1, and BARD1 (Fig. 1b and Supplementary Data 1). Intriguingly, we also reproducibly recovered AUNIP/C1orf135[42], a largely uncharacterized protein, as a putative CtIP-interacting protein (Fig. 1b and Supplementary Data 1). To determine if AUNIP could indeed form a complex with CtIP, we conducted reverse TAP experimentations using a cell line engineered to stably express SFB-tagged AUNIP, and uncovered CtIP, MRE11, RAD50, and DNA2 as major AUNIP-associated proteins (Fig. 1c, d and Supplementary Data 2). These findings strongly suggest that AUNIP and CtIP exist in the same protein complex in cells. AUNIP was originally identified as an Aurora-A-interacting protein and has been shown to target Aurora-A to spindle poles[42]. Interestingly, AUNIP does not contain any known functional motifs, and human AUNIP orthologs are found exclusively in vertebrates, such as *Pan troglodytes*, *Bos Taurus*, *Mus musculus*, *Canis lupus familiaris*, and *Xenopus laevis*.

To verify the TAP-mass spectrometry data, we first performed co-immunoprecipitation experiments using lysates derived from HEK293T cells transiently transfected with plasmids that encode SFB-tagged AUNIP together with expression constructs that encode Myc-tagged CtIP, -MRE11, -RAD50, or -NBS1. As shown in Supplementary Fig. 1a, AUNIP interacted with CtIP and all three components of the MRN complex, but not with the unrelated protein Morc3. We further carried out reciprocal co-immunoprecipitation experiments in HeLa cells and confirmed

that endogenous AUNIP and CtIP proteins formed a complex in vivo (Fig. 1e). The complex formation was not affected by CPT treatment, and was resistant to benzonase treatment, arguing against the possibility that the observed interaction was mediated by DNA/RNA (Fig. 1e). Notably, cross-species amino-acid sequence alignment revealed that both termini of the AUNIP peptide are highly conserved (Supplementary Fig. 1b). We thus tested whether these evolutionarily-conserved regions of AUNIP are involved in its interaction with CtIP. Co-immunoprecipitation experiments showed that AUNIP interacted with CtIP through its conserved C-terminus, since a deletion mutant lacking the C-terminal 77 amino acids (1–280) was unable to co-precipitate with CtIP (Fig. 1f).

To test whether the interaction between AUNIP and CtIP was direct, we performed GST pull-down assays using recombinant GST-tagged AUNIP and MBP-tagged CtIP purified from *E. coli*. As shown in Fig. 1g, wild-type, but not the 1–280 deletion mutant of AUNIP, directly interacted with CtIP in vitro. Interestingly, the C-terminal region alone (281–357) was also able to directly interact with CtIP (Fig. 1g). Altogether, these results suggested that the conserved C-terminal region of AUNIP is necessary and sufficient for its interaction with CtIP.

**AUNIP accumulates at sites of DNA damage**. Given that AUNIP exists in a complex with CtIP, we examined whether AUNIP may also be recruited to damaged replication forks or to DNA lesions. Because we were unable to detect AUNIP by indirect immunofluorescence experiments using our in-house anti-AUNIP antibodies, we, therefore, utilized the CRISPR/Cas9-directed recombinant adeno-associated virus (rAAV)-mediated gene targeting approach[43] to integrate an SFB tag onto the C-terminus of the endogenous AUNIP gene (Supplementary Fig. 2a). DNA sequencing results confirmed that the tag was targeted correctly, and expression of the AUNIP-SFB protein was confirmed on immunoblots probed with anti-Flag antibody (Supplementary Fig. 2b). As shown in Fig. 2a, in most untreated cells, AUNIP showed a diffuse staining pattern. However, when cells were treated with the topoisomerase I inhibitor CPT, AUNIP concentrated into distinct nuclear foci (Fig. 2a). Notably, although AUNIP foci only partially colocalized with RPA2 foci (Supplementary Fig. 2c), they were found exclusively in RPA2 foci-positive cells (Supplementary Fig. 2d), indicating that AUNIP foci may correspond to perturbed replication forks.

We next laser micro-irradiated U2OS cells that stably express GFP-tagged AUNIP to determine whether AUNIP is recruited to sites of DNA damage. As shown in Fig. 2b, GFP-AUNIP was readily recruited to RPA2- and γH2AX-marked laser-generated stripes. Moreover, similar to GFP-AUNIP, endogenous AUNIP can also be detected at laser micro-irradiated tracks (Fig. 2c).

To determine which domain is responsible for the recruitment of AUNIP to laser-irradiated sites, we generated a series of AUNIP deletion mutants that span the entire AUNIP protein (Fig. 2d), and monitored the recruitment of each of these deletion mutants. Since the very end of the N-terminus of AUNIP contained a functional nuclear localization signal (NLS) (K[21]RRK[24]) (Supplementary Fig. 2e), we added an NLS to the N-terminus of the 81–357 deletion mutant (lacking the N-terminal 80 amino acids) to ensure its proper nuclear localization. As shown in Fig. 2e, aside the NLS-81–357 deletion mutant, all other AUNIP mutants retained the ability to accumulate at laser-induced γH2AX-marked DNA damage tracks, indicating that the N-terminus of AUNIP is indispensable for its recruitment to DNA lesions. Further deletion analysis narrowed down the region essential for AUNIP accumulation at DNA damage tracks to residues 25–70 (Fig. 2e).

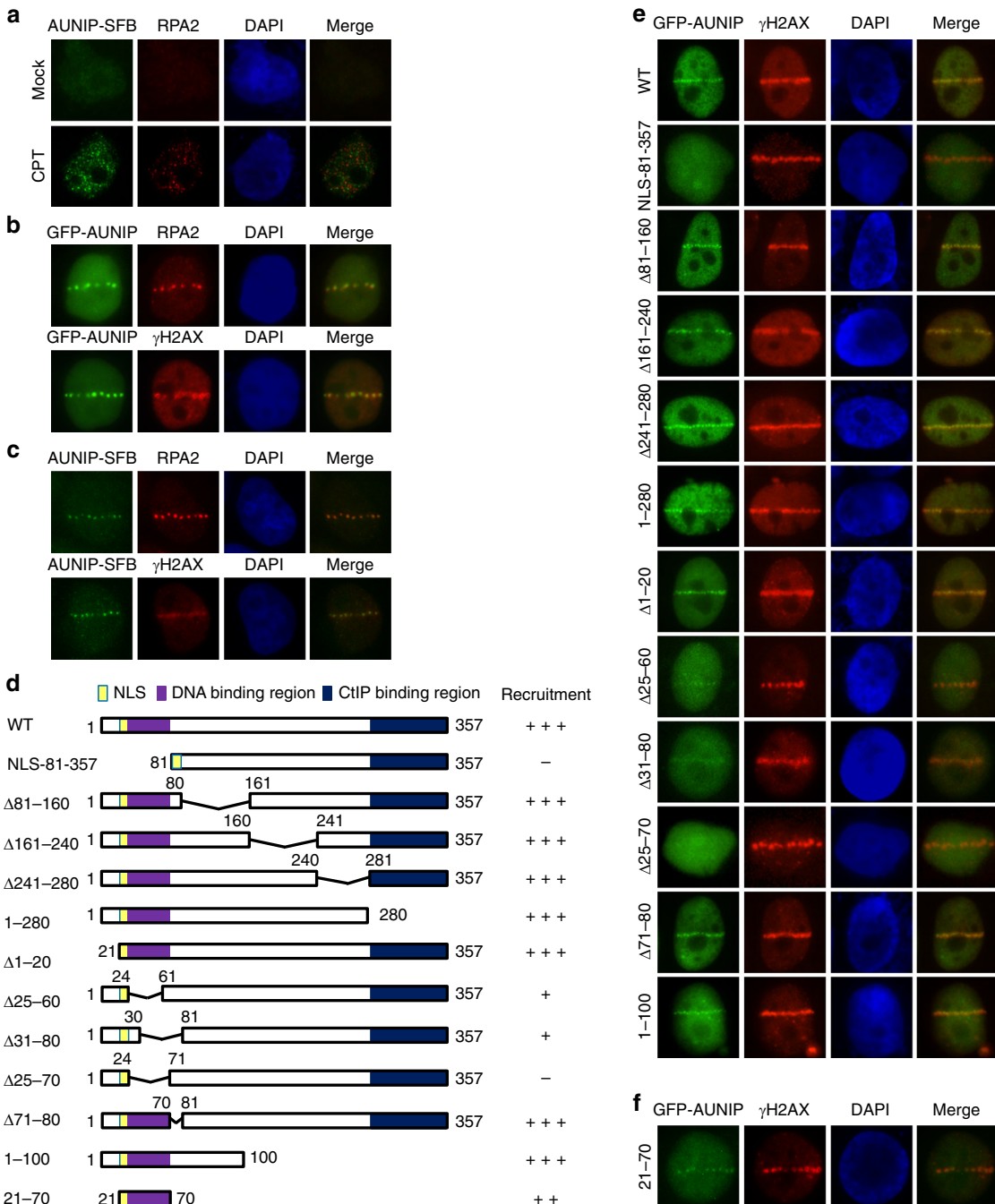

**Fig. 2** AUNIP accumulates at perturbed replication forks and at sites of DNA damage. **a** Endogenous AUNIP accumulates at perturbed replication forks. AUNIP-SFB knock-in HeLa cells were either mock treated or treated with 1 μM CPT for 6 h before they were processed for indirect immunofluorescent analysis for AUNIP-SFB (α-Flag) and RPA2 (α-RPA2). **b** GFP-tagged AUNIP is recruited to laser-induced DNA damage sites. U2OS cells transfected with GFP-tagged AUNIP were laser micro-irradiated. After 10 min, cells were stained with anti-RPA2 or anti-γH2AX antibody. **c** AUNIP-SFB knock-in HeLa cells were laser micro-irradiated. Ten minutes post irradiation, cells were processed for immunostaining experiments using anti-Flag and anti-RPA2/γH2AX antibodies. **d** Schematic representation of AUNIP mutants used in this study. **e** Amino acids 25–70 of AUNIP is required for its recruitment to sites of DNA damage. Cells transfected with GFP-tagged wild-type AUNIP or its mutants were laser micro-irradiated. Cells were processed for immunostaining experiments using anti-γH2AX antibody. **f** Amino acids 21–70 of AUNIP is sufficient for its recruitment to sites of DNA damage. Cells transfected with GFP-AUNIP-21–70 were laser micro-irradiated. Cells were labeled with anti-γH2AX antibody to visualize DNA damage tracks

To further clarify whether the region that encompasses amino acids 25–70 of AUNIP alone is sufficient to localize AUNIP to sites of DNA damage, we generated a construct encoding this region together with a NLS (GFP-AUNIP-21–70) (Fig. 2d). As shown in Fig. 2f, GFP-AUNIP-21–70 was able to localize to laser micro-irradiated sites.

**AUNIP promotes HR and inhibits NHEJ.** HR and non-homologous end-joining (NHEJ) are the two major DSB repair pathways in eukaryotic cells[2–6]. The observation that AUNIP existed in a complex with CtIP and accumulated at DNA damage sites prompted us to test whether AUNIP may be involved in DSB repair. To this end, we used two previously established U2OS cell

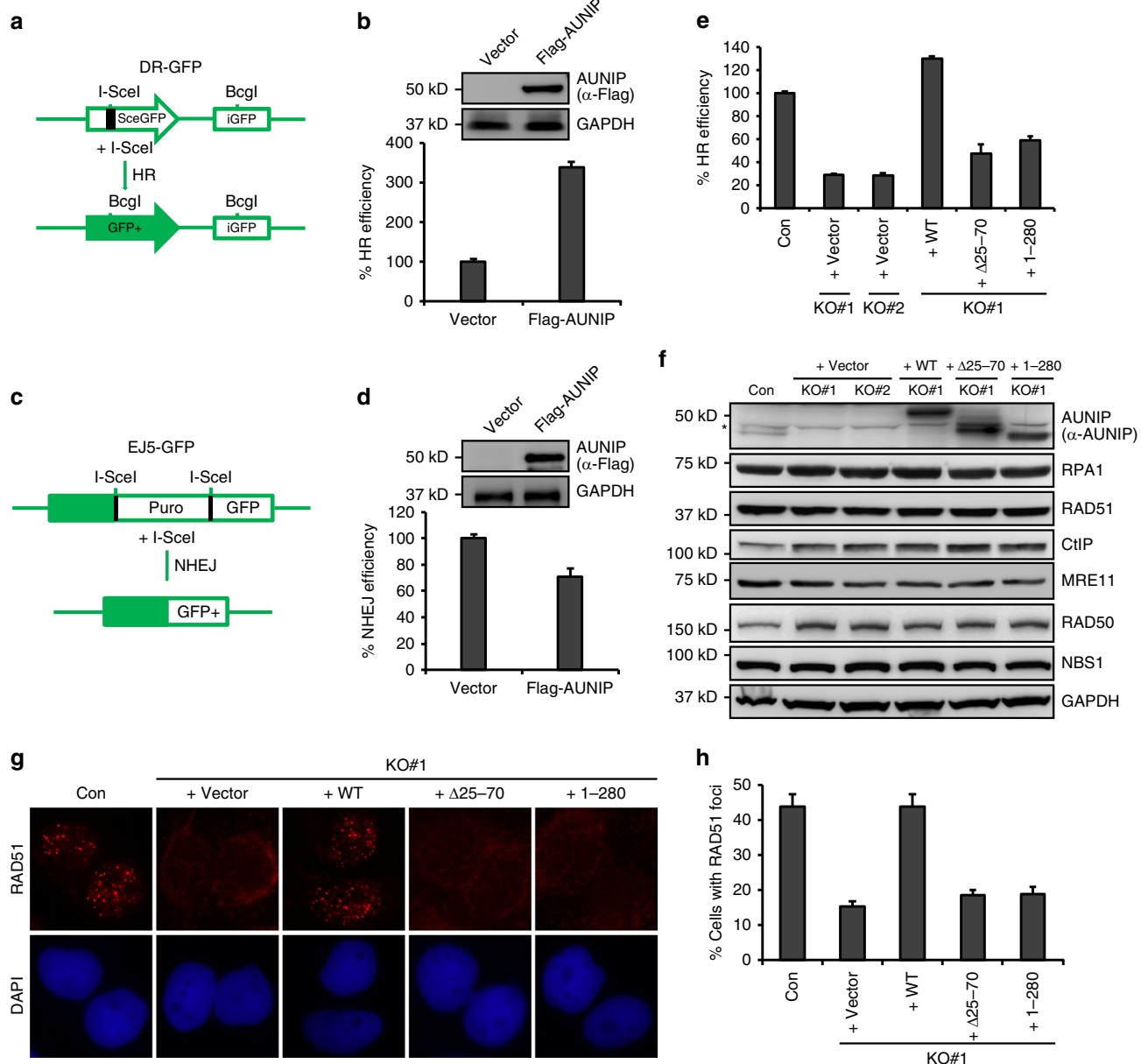

**Fig. 3** AUNIP promotes HR and inhibits NHEJ. **a** Schematic illustration of the GFP-based HR reporter assay. **b** AUNIP overexpression promotes HR. U2OS DR-GFP cells stably expressing Flag-tagged AUNIP were electroporated with pCBASce construct and were assayed for HR efficiency by monitoring GFP positivity 48 h post electroporation. Data represent mean ± SEM from three independent experiments. **c** Schematic illustration of the GFP-based NHEJ reporter assay. **d** AUNIP overexpression inhibits NHEJ. U2OS EJ5-GFP cells stably expressing Flag-tagged AUNIP were electroporated with pCBASce construct and were assayed for NHEJ efficiency by monitoring GFP positivity 48 h post electroporation. Data represent mean ± SEM from three independent experiments. **e** AUNIP knockout impairs HR. AUNIP-deficient U2OS DR-GFP cells stably expressing an empty vector (Vector), Flag-tagged wild-type AUNIP, the Δ25–70 mutant, or the 1–280 mutant were electroporated with pCBASce construct for 48 h before they were assayed for HR efficiency. Data represent mean ± SEM from three independent experiments. **f** The overall levels of a panel of key HR proteins are not markedly affected by AUNIP silencing. Whole-cell lysates from indicated cells were subjected to western blot analysis using indicated antibodies. Asterisks indicate nonspecific bands. **g**, **h** Knockout of AUNIP impairs CPT-induced RAD51 foci formation. AUNIP-deficient cells stably expressing an empty vector (Vector), Flag-tagged wild-type AUNIP, the Δ25–70 mutant, or the 1–280 mutant were treated with 1 μM CPT for 6 h before being processed for RAD51 immunofluorescence. Representative RAD51 foci are shown in **g**. Data represent mean ± SEM from three independent experiments (**h**)

lines that harbor a chromosomally-integrated copy of the DR-GFP[44] or EJ5-GFP[45] reporter to measure the repair rates of I-SceI-induced DSBs in chromosomal DNA by HR or NHEJ, respectively (Fig. 3a, c). We first stably overexpressed AUNIP in these reporter cell lines and measured the efficiency of DSB repair. As shown in Fig. 3b, overexpression of AUNIP led to a dramatic increase in the frequency of HR. Conversely, overexpression of AUNIP substantially inhibited NHEJ (Fig. 3d).

These results suggest that AUNIP promotes HR and inhibits NHEJ.

To verify and extend the above findings, we took advantage of the CRISPR/Cas9 gene-editing approach to inactivate AUNIP gene expression in the U2OS DR-GFP cell line. We confirmed the absence of AUNIP in this cell line by Western blotting analysis (Supplementary Fig. 3a). Consistent with the notion that overexpression of AUNIP promotes HR and inhibits NHEJ,

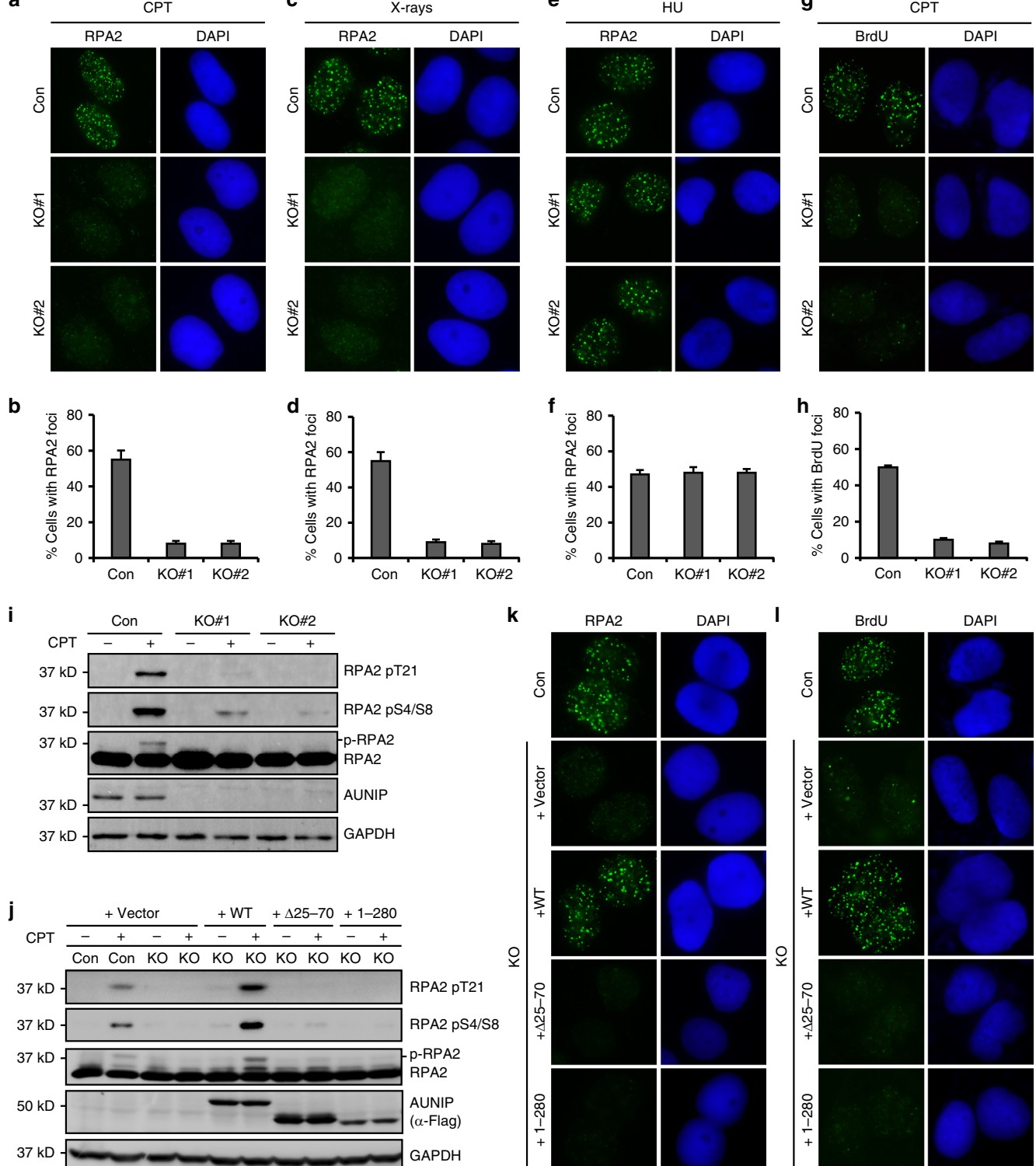

**Fig. 4** AUNIP stimulates DNA end resection. **a**–**d** Knockout of AUNIP impairs CPT-or X-ray-induced RPA2 foci formation. AUNIP-deficient HeLa cells were treated with 1 μM CPT or 10 Gy X-rays and were processed for RPA2 immunostaining experiment 1 h after. Representative RPA2 foci are shown in **a**, **c**. Data represent mean ± SEM from three independent experiments (**b**, **d**). **e**, **f** AUNIP does not affect HU-induced RPA2 foci formation. AUNIP-deficient HeLa cells were treated with 10 mM HU for 1 h before being processed for immunofluorescence studies using anti-RPA2 antibody. Representative RPA2 foci are shown in **e**. Data represent mean ± SEM from three independent experiments **f**. (**g**, **h**) Knockout of AUNIP impairs ssDNA formation. Wild-type or AUNIP-deficient U2OS cells were labeled with BrdU for 24 h, and were subsequently treated with 1 μM CPT for 1 h. Cells were then stained with anti-BrdU antibody under non-denaturing conditions. Representative BrdU foci are shown in **g**. Data represent mean ± SEM from three independent experiments (**h**). **i** Knockout of AUNIP impairs CPT-induced RPA2 phosphorylation. Cells were treated with 1 μM CPT for 1 h and were processed for western blot analysis. **j**–**l** The AUNIP mutants lacking the ability to either localize to DNA damage sites or to bind to CtIP failed to restore RPA2 phosphorylation and RPA2/BrdU foci formation in AUNIP-deficient cells. AUNIP-deficient cells stably expressing vector control (Vector), wild-type AUNIP, the Δ25–70 mutant, or the 1–280 mutant were treated with 1 μM CPT for 1 h. Cells were subsequently processed for western blot analysis (**j**) or for indirect immunofluorescence studies using anti-RPA2 (**k**) or anti-BrdU antibody (**l**)

knockout of AUNIP led to a dramatic decrease in the frequency of HR (Fig. 3e, f). In addition, downregulation of AUNIP increased the frequency of NHEJ (Supplementary Fig. 3b, c). Notably, these effects were not a simple consequence of the changes in cell cycle phase distribution (Supplementary Fig. 3d).

To investigate how AUNIP facilitates the HR process, we examined whether AUNIP deficiency may affect RAD51 foci formation in response to CPT. As shown in Fig. 3g, h, in the absence of AUNIP, CPT-induced RAD51 foci formation were severely impaired. By contrast, AUNIP silencing had no effect on γH2AX and 53BP1 foci formation (Supplementary Fig. 3e–h). Moreover, expression of RAD51 and relevant HR factors were not affected by loss of AUNIP (Fig. 3f). To ensure that the observed phenotypes were directly related to AUNIP inactivation, we reconstituted AUNIP-deficient cells with wild-type AUNIP and performed rescue experiments. As shown in Fig. 3e–h, re-expression of wild-type AUNIP in AUNIP-deficient cells fully restored RAD51 focus formation and HR repair, strongly indicating that these observed HR defects are associated with AUNIP deficiency. By contrast, re-introduction of AUNIP mutants that were either defective in accumulating at DNA damage sites or in interacting with CtIP did not reverse the AUNIP-associated deficits in HR repair repair (Fig. 3e–h). These results suggest that the ability of AUNIP to localize to DSBs and to bind to CtIP are both critical for its function in DSB repair.

**AUNIP stimulates DNA end resection**. The key event that controls the DSB repair pathway choice is DNA end resection, which prevents repair by NHEJ and commits cells to homology-dependent repair[25]. Because AUNIP promotes HR and inhibits NHEJ, we speculated that AUNIP might regulate DNA end resection. We, therefore, determined whether knockout of AUNIP in HeLa cells would affect RPA2 foci formation in response to CPT or X-ray treatment. As shown in Fig. 4a–d, RPA2 foci formation was markedly impaired in the absence of AUNIP. Similar results were obtained in U2OS cells (Supplementary Fig. 4a–e). By contrast, AUNIP silencing had no obvious effect on RPA2 foci formation induced by the replication inhibitor hydroxyurea (HU), a drug that can generate resection-independent exposure of ssDNAs by uncoupling the DNA polymerase and the MCM helicase at replication forks (Fig. 4e, f). These results suggest that the loss of AUNIP specifically impaired RPA2 foci formation in response to DNA damage. To further substantiate the role of AUNIP in DNA end resection, we used a BrdU-staining method to monitor ssDNA levels under non-denaturing conditions. As shown in Fig. 4g, h, BrdU foci formation was dramatically reduced in AUNIP-deficient cells, indicating that AUNIP is required for the efficient generation of RPA-coated ssDNA at resected DSBs.

The RPA subunit RPA2 becomes hyper-phosphorylated in cells when bound to ssDNAs and the inhibition of key resection factors leads to a decrease in this mark. In line with a proposed role of AUNIP in stimulating DNA end resection, its deficiency severely impaired CPT-induced RPA2 hyper-phosphorylation on Ser-4/Ser-8 and Thr-21 (Fig. 4i). Importantly, defects in RPA2 and BrdU foci formation, as well as dampened RPA2 hyper-phosphorylation, were fully restored by re-expression of wild-type AUNIP in AUNIP-deficient cells, but not by mutants that either lack the ability to localize to sites of DNA damage or bind to CtIP (Fig. 4j–l and Supplementary Fig. 4f, g). Moreover, re-expression of the AUNIP mutant that lacks the NLS (ΔKRKK) was also unable to reverse these defects (Supplementary Fig. 4h–j). Taken together, these results indicate that AUNIP may control DSB repair pathway choice by promoting DNA end resection.

**AUNIP promotes CtIP recruitment to sites of DNA damage**. To obtain mechanistic insight into how AUNIP contributes to DNA end resection, we assessed whether AUNIP may affect DNA damage-dependent recruitment of factors known to be involved in this process. To this end, we first performed time-lapse imaging of micro-irradiated U2OS cells that stably express GFP-tagged AUNIP, -CtIP or -NBS1 (the GFP tag was added to the C-terminus of NBS1 to avoid unwanted functional deficits) to compare the recruitment kinetics of AUNIP with that of CtIP and NBS1. As shown in Fig. 5a, b, the recruitment of NBS1 to sites of DNA damage could be observed as early as 25 s after micro-irradiation. By contrast, we observed a substantial delay in the recruitment of GFP-CtIP to laser-induced DNA lesions, compared with that of NBS1-GFP (Fig. 5a, b). These findings are in agreement with previous observations where the MRN complex is required for efficient recruitment of CtIP to sites of DNA damage[19, 46, 47]. Remarkably, GFP-AUNIP accumulated at sites of laser-inflicted DNA damage tracks with kinetics similar to NBS1-GFP, indicating that AUNIP might also act upstream of CtIP in DNA end resection (Fig. 5a, b). Indeed, loss of AUNIP significantly and reproducibly hampered GFP-CtIP recruitment to laser-generated DSB tracks (Fig. 5c, d). Consistently, loss of AUNIP also impaired the recruitment of CtIP to CPT-induced DNA damage foci (Fig. 5e, f). More importantly, wild-type AUNIP, but not its mutants that either lack the ability to localize to sites of DNA damage or in binding to CtIP, completely restored CtIP recruitment to DNA damage sites in AUNIP-deficient cells (Fig. 5c–f). These results, taken together with the observation that CtIP depletion did not noticeably affect AUNIP accumulation at sites of laser-induced DNA damage (Fig. 5g–i), argue in favor of the idea that CtIP functions downstream of both MRN and AUNIP in DNA end resection.

We next examined whether the localization of AUNIP to sites of DNA damage is dependent on the MRN complex or vice versa. As shown in Supplementary Fig. 5a–c, AUNIP deficiency did not affect the accumulation of NBS1-GFP at sites of laser-inflicted DNA damage in both the presence and absence of H2AX. Similarly, depletion of NBS1 had no significant effect on the recruitment of AUNIP to sites of laser-induced DNA damage (Fig. 5g–i). These results suggest that, despite similar recruitment kinetics, DNA damage-induced recruitment of AUNIP and the MRN complex may be regulated independently.

**AUNIP is a structure-specific DNA-binding protein**. Having established that AUNIP accumulates at DNA lesions and facilitates CtIP-dependent DNA-end resection, we next sought to determine the mechanism by which AUNIP is recruited to sites of DNA damage. We first examined whether AUNIP recruitment to DNA damage sites may be dependent on the ATM or ATR kinase pathway. As shown in Supplementary Fig. 6a, ATM inhibition did not noticeably affect AUNIP recruitment to sites of DNA damage. In line with this notion, depletion of H2AX or RNF8 had no effect on AUNIP concentration at DNA breaks (Supplementary Fig. 6b, c). Moreover, there was also no significant change in the localization of AUNIP to sites of DNA damage using an ATR inhibitor (Supplementary Fig. 6a). These findings prompted us to hypothesize that AUNIP might be recruited to sites of DNA damage via direct interaction with DNA. To test this hypothesis, recombinant wild-type AUNIP was purified and analyzed in electromobility shift assays against a variety of DNA substrates. As shown in Fig. 6a–i, AUNIP bound strongly to splayed arm, 5′-flap, and 3′-flap DNA substrates, but only weakly to double-stranded DNA (dsDNA), dsDNA with a tail, or ssDNA. Notably, the Δ25–70 deletion mutant, which had lost the ability to accumulate at DSB sites, failed to bind to splayed arm under the same

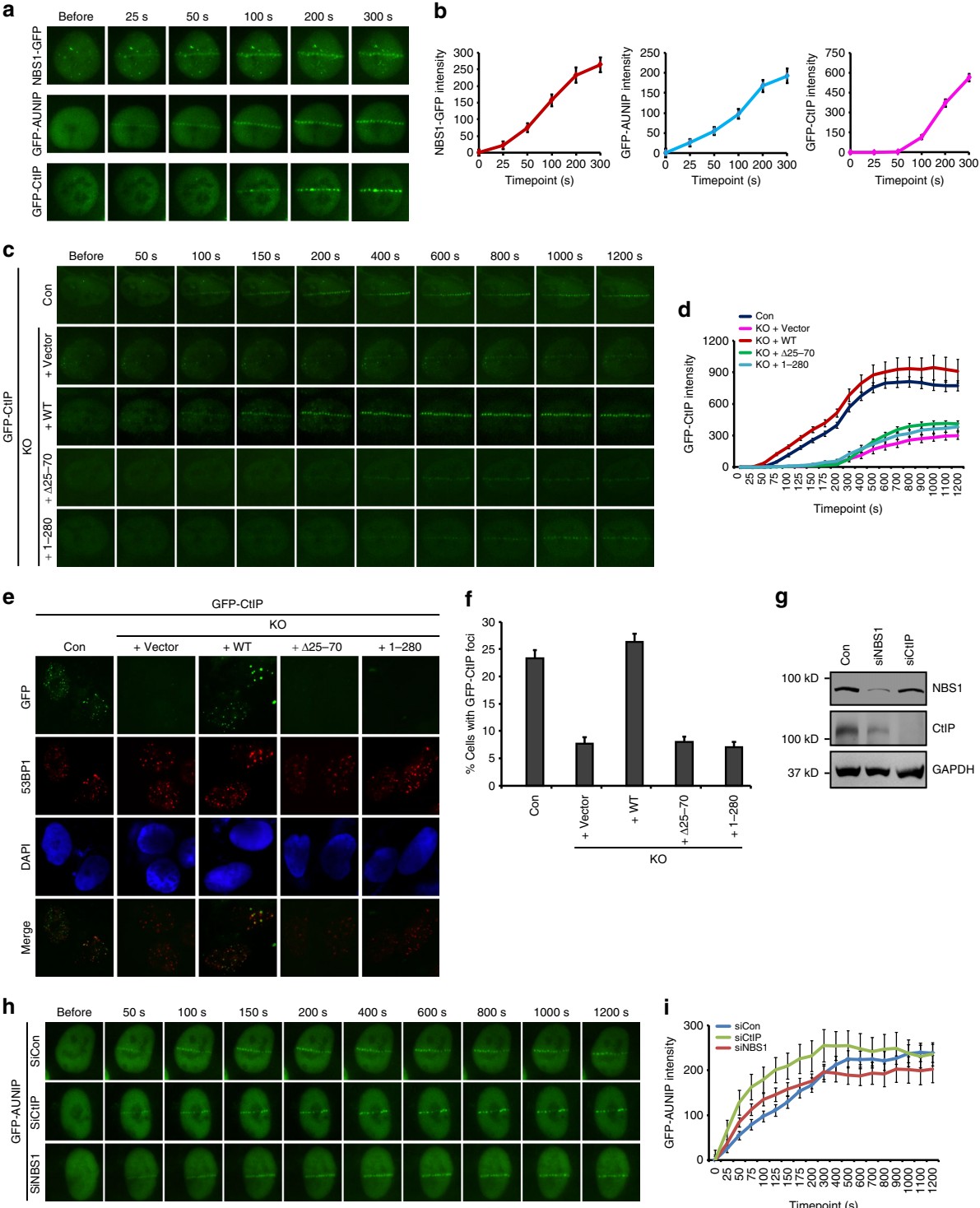

**Fig. 5** AUNIP promotes CtIP recruitment to DNA damage sites. **a**, **b** Kinetic studies of accumulation of GFP-tagged AUNIP, NBS1, or CtIP at laser-induced DNA damage sites. U2OS cells stably expressing GFP-AUNIP, NBS1-GFP, or GFP-CtIP were laser micro-irradiated and were monitored by live cell imaging (**a**). The intensity of fluorescence at the site of damage was quantified **b**. Data were derived from analysis of at least 20 cells in each experiment and are presented as mean ± SEM. **c**, **d** Knockout of AUNIP impairs CtIP recruitment to laser-induced DNA damage sites. AUNIP-deficient U2OS cells stably expressing empty vector (Vector), wild-type AUNIP, the Δ25–70 mutant, or the 1–280 mutant were infected with a lentiviral vector expressing GFP-tagged CtIP. After 48 h, cells were laser micro-irradiated and were monitored by live cell imaging (**c**). Data were derived from analysis of at least 20 cells in each experiment and are presented as mean ± SEM (**d**). **e**, **f** Knockout of AUNIP impairs CPT-induced CtIP foci formation. Cells infected with a lentiviral vector encoding GFP-tagged CtIP were treated with 1 μM CPT for 1 h and were processed for immunofluorescence studies. Representative CtIP foci are shown in **e**. Data represent mean ± SEM from three independent experiments (**f**). **g–i** Neither NBS1 nor CtIP is required for AUNIP damage recruitment. Cells transfected with indicated siRNAs were laser micro-irradiated and were monitored by live cell imaging (**h**). Data were derived from analysis of at least 20 cells in each experiment and are presented as mean ± SEM (**i**). Knockdown efficiency of NBS1/CtIP was confirmed by western blotting (**g**)

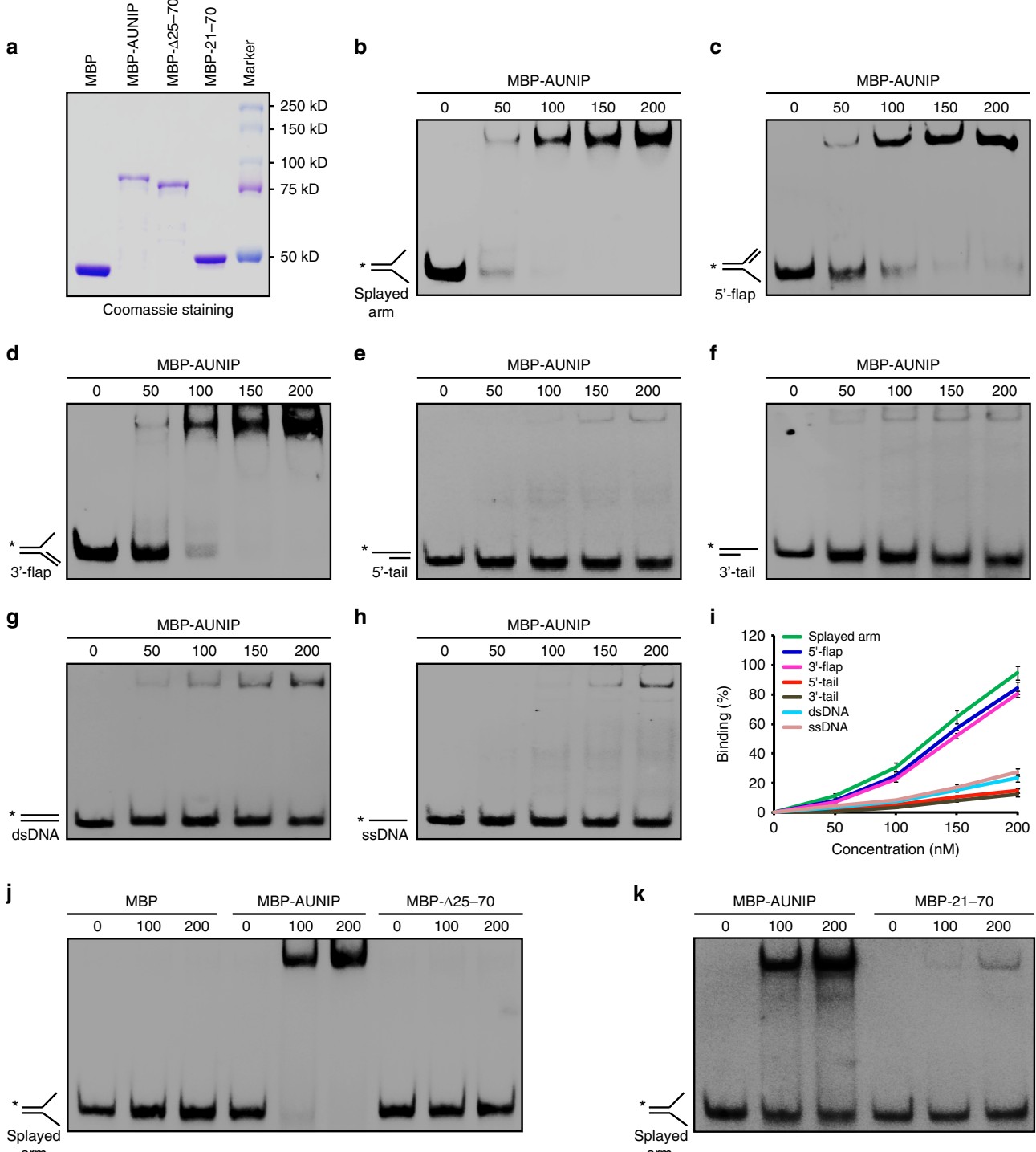

**Fig. 6** DNA-binding activity of AUNIP. **a** SDS-PAGE profile of purified MBP-tagged wild-type and mutants of AUNIP. **b–h** DNA-binding activity of AUNIP. Biotin-labeled DNA substrates (20 nM each) were incubated with increasing amounts of purified MBP-AUNIP (0, 50, 100, 150, 200 nM). **i** Quantification of the results is shown in **b–h**. Data were derived from three independent experiments and are presented as mean ± SEM. **j** The region encompassing amino acids 25–70 of AUNIP is required for its DNA-binding activity. Biotin-labeled DNA substrates (20 nM) were incubated with increasing amounts of purified MBP-tagged AUNIP or its mutant (0, 100, 200 nM). **k** The region encompassing amino acids 21–70 of AUNIP is sufficient for DNA binding. Biotin-labeled DNA substrates (20 nM) were incubated with increasing amounts of purified MBP-AUNIP or its mutant (0, 100, 200 nM)

experimental conditions(Fig. 6j). More importantly, the fragment encompassing residues 21–70, which is sufficient to target AUNIP to sites of DNA damage, exhibited the ability to bind to splayed arm, albeit with lower affinity (Fig. 6k). Taken together, these results suggest that AUNIP is a structure-specific DNA-binding protein with a preference for DNA substrates that mimic damaged replication forks, and that this DNA-binding activity is essential for its ability to translocate to sites of DNA damage in cells.

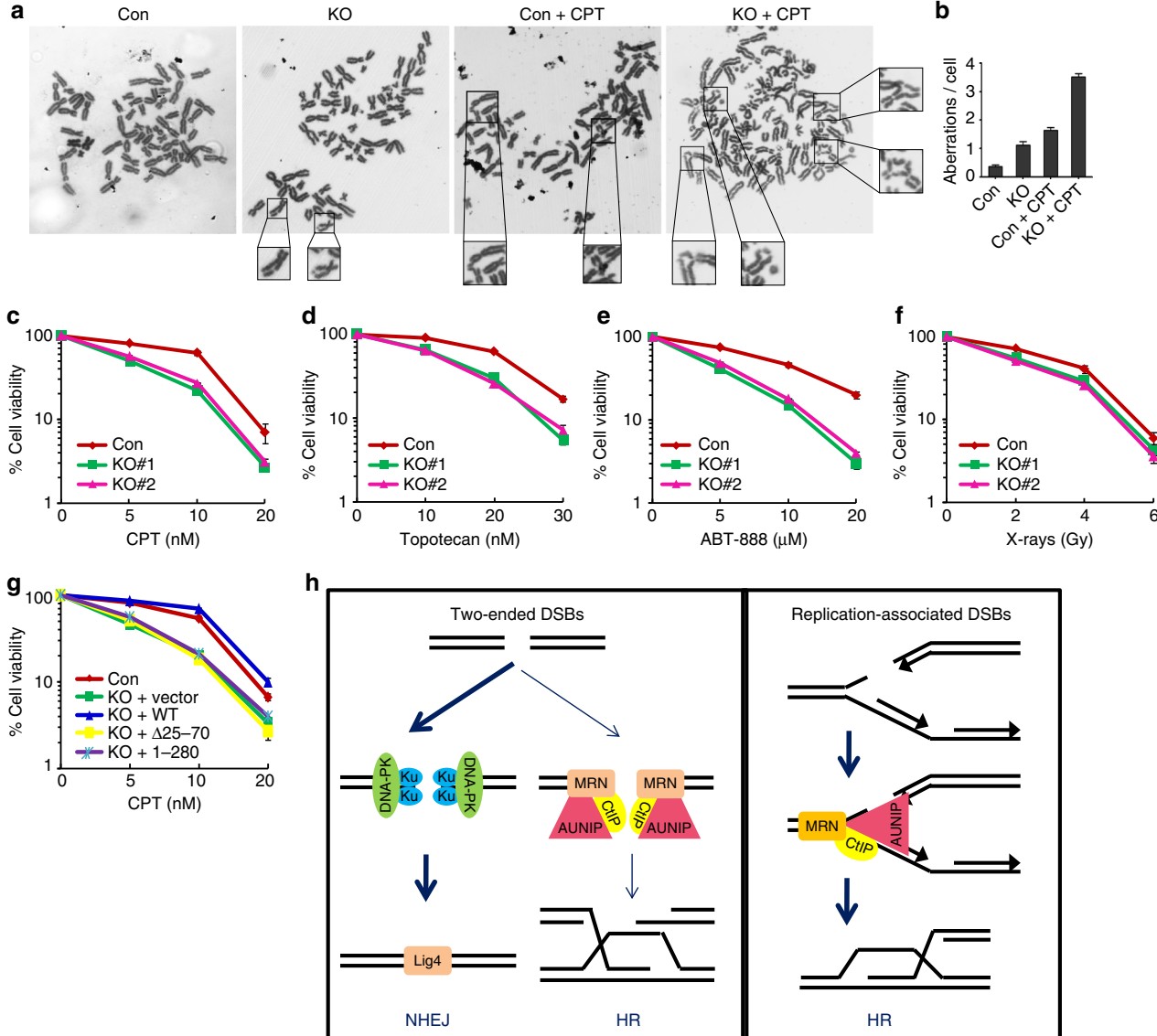

**Fig. 7** AUNIP is required for genome stability and cellular resistance to DSB-inducing agents. **a** Wild-type or AUNIP-deficient HeLa cells were either mock treated or treated with 40 μM CPT for 12 h. Metaphase spreads were then prepared following standard procedures. Representative images are shown in **a**. Arrows indicate chromosome aberrations. Quantification of chromosomal aberrations from indicated cells is shown in **b**. Data represent the average for two independent experiments. At least 50 metaphases were counted in each experiment. **c–f** Sensitivity of wild-type cells or AUNIP-deficient HeLa cells to CPT, Topotecan, ABT-888, or IR was determined by a colony formation assay. Data represent means ± SEM from three independent experiments. **g** CPT sensitivity of wild-type or AUNIP-deficient cells that stably express empty vector (Vector), wild-type AUNIP, the Δ25–70 mutant, or the 1–280 mutant was determined by a colony formation assay. Data represent means ± SEM from three independent experiments. **h** Working model depicting a proposed role of AUNIP in DSB repair choice. Left panel: Noting that AUNIP displays low affinities for dsDNA ends, most of the two-ended DSBs are recognized and are repaired by the NHEJ machinery. Right panel: AUNIP enhances CtIP tethering at perturbed replication forks and directs replication-associated one-ended DSBs toward the HR repair pathway

**AUNIP confers resistance to DSB-inducing agents**. Given that AUNIP is required for efficient DNA end resection and HR repair, we speculated that AUNIP deficiency may cause genome instability and hypersensitize cells to DSB-inducing agents, in particular to agents that induce replication-associated DSBs. Indeed, cells deficient in AUNIP exhibited a significant increase in chromosomal aberrations as determined by metaphase chromosome analyses (Fig. 7a, b). Moreover, loss of AUNIP resulted in cell hypersensitivity to agents that induced replication-associated DSBs, including CPT, Topotecan (a CPT derivative, also known as CPT-11), and the PARP inhibitor ABT-888 (Fig. 7c–e). By contrast, AUNIP knockout cells exhibited only mild sensitivity to X-rays (Fig. 7f), consistent with the notion that

X-ray-induced DSBs are predominantly repaired by NHEJ[33, 35, 36]. Importantly, re-expression of wild-type AUNIP, but not the mutants defective in DNA or CtIP binding, restored CPT resistance in AUNIP-deficient cells, indicating that both the CtIP-binding domain and the DNA-binding activity of AUNIP are critical for its function in promoting cellular resistance to DSB-inducing agents (Fig. 7g).

**Discussion**
The key event that controls DSB repair pathway choice is DNA end resection, which prevents repair by NHEJ and commits cells to homology-dependent repair pathways. In this study, we have

provided several lines of evidence to show that AUNIP is a critical regulator of DNA end resection. First, overexpression of AUNIP promoted HR and inhibited NHEJ. Consistently, AUNIP silencing decreased the frequency of HR but increased the frequency of NHEJ. Second, AUNIP physically interacted with CtIP and was required for efficient CtIP recruitment to sites of DNA damage and the subsequent CtIP-dependent DNA end resection. Third, AUNIP possessed intrinsic DNA-binding activity, a feature that was required for its relocalization to sites of DNA damage and its function in DNA end resection. Finally, AUNIP-deficient cells showed a marked increase in cellular sensitivity to DSB-inducing agents, particularly to those that induce replication-associated DSBs. Our results provide novel insights into how DSBs are recognized and may be channeled to specific repair pathways.

A scenario is emerging in which the nature of DSBs caused by stalled replication forks is distinct from that caused by X- or γ-rays. Indeed, ample evidence now indicate that DSBs arising from stalled replication forks are predominantly repaired by HR, whereas those that are induced by X- or γ-rays are mostly processed by NHEJ[33, 35, 36]. However, the manner in which these replication-associated DSBs are distinguished from two-ended DSBs in cells is largely unclear.

In this study, we found that AUNIP preferentially bound to DNA substrates that mimicked structures generated at stalled replication forks and was critical for CtIP recruitment to sites of DNA damage. We proposed that AUNIP may serve as a molecular sensor for stalled replication forks, and by tethering CtIP at DNA intermediates generated during replication fork stalling, directs replication-associated DSBs toward the HR repair pathway (Fig. 7h). On the other hand, since AUNIP displayed low affinities for dsDNA ends, it is tempting to speculate that the CtIP protein may not be fully activated at two-ended DSBs, including those that are generated by X- or γ-rays (Fig. 7h). This model thus offers an explanation to address and to further experimentally examine why two-ended DSBs are primarily repaired by NHEJ. Importantly, this model is supported by our observations where AUNIP-deficient cells were hypersensitive to genotoxins that induce replication-associated DSBs, but were only mildly sensitive to X-rays. Thus, the identification of AUNIP represents an important step toward understanding how cells channel repair processes to resolve specific problems associated with DNA damage.

CtIP and its homologs are well-known molecular switches that regulate DNA end resection, and their function in this process is strictly dependent on their appropriate recruitment to sites of DNA damage. However, exactly how CtIP is recruited to DSBs is not fully understood. Studies have shown that the MRN complex physically interacts with CtIP and helps retain CtIP at DSBs[19, 46, 47]. In addition to its direct involvement in CtIP recruitment, the MRN complex also facilitates CtIP recruitment through tethering broken DNA ends and in promoting ATM activation[19, 46]. It has been suggested that ATM-mediated CtIP phosphorylation may unmask the DNA-binding motif in CtIP, which then binds to DNA at DSBs, resulting in CtIP recruitment to DSBs[19]. In resemblance to the MRN complex, AUNIP also binds to DNA, forms a complex with CtIP, and is required for efficient CtIP recruitment to sites of DNA damage. However, AUNIP is not involved in ATM activation (Supplementary Fig. 7). Moreover, DNA damage-induced recruitment of AUNIP and the MRN complex are independent of each other. These results indicate that AUNIP may serve as a DNA-binding module for the AUNIP-CtIP-MRN complex to target specific DNA lesions, and that the multiple DNA-binding modules on the AUNIP-CtIP-MRN complex might play coordinated roles to support full activation of CtIP at sites of DNA damage. In addition to AUNIP, the MRN complex, and ATM, several other proteins including BRCA1, LEDGF/p75, SRCAP, and USP4, are also involved directly or indirectly in recruiting CtIP to sites of DNA damage[48–52]. However, the mechanistic details with which these proteins cooperate to support optimal recruitment of CtIP is still not clear and warrants further investigation.

In summary, the identification and biochemical characterization of the AUNIP protein described here provides new insights into the molecular mechanism by which DSBs, in particular replication-associated DSBs, may be recognized and repaired in human cells. Understanding the detailed mechanisms and regulation of this process will help in the design of more efficient anti-cancer therapeutics.

## Methods

**Antibodies**. Polyclonal anti-AUNIP antibodies (1:200 dilution) were generated by immunizing rabbits with MBP-AUNIP (residues 81–357) or MBP-AUNIP (residues 220–357) fusion proteins expressed and purified from *E. Coli*. Polyclonal anti-RAD51 (1:2000 dilution) and anti-RNF8 (1:1000 dilution) antibodies were generated by immunizing rabbits with GST-RAD51 (residues 1–339), or GST-RNF8 (residues 1–485) fusion proteins expressed and purified from *E. Coli*, respectively. Antisera were affinity-purified against the immunized antigens using the Amino-Link plus Immobilization and purification kit (Pierce). Polyclonal anti-phospho-RPA2 (S4/S8) (A300-245A, 1:3000 dilution) and anti-NBS1 (A301-284A, 1:1000 dilution) antibodies were purchased from Bethyl Laboratories. Anti-CtIP (61141, Clone: 14-1, 1:1000 dilution), anti-MRE11 (GTX70212, 1:1000 dilution), and anti-Myc (9E10, 1:1000 dilution) antibodies were purchased from Active Motif, Gene Tex, and Covance, respectively. Anti-RPA2 (ab2175, 1:2000 dilution), anti-RAD50 (ab89, 1:1000 dilution), anti-ATM (ab199726, 1:1000 dilution), anti-53BP1 (ab175188, 1:100 dilution), anti-H2AX (ab11175, 1:2000 dilution), and anti-phospho-ATM (S1981) (ab81292, 1:1000 dilution) antibodies were purchased from Abcam. Anti-GAPDH (MAB374, 1:5000 dilution) and anti-phospho-H2AX (Ser319) (Clone JBW301, 1:1000 dilution) antibodies were purchased from Millipore. Anti-Flag (Clone M2, 1:10000 dilution) antibody was purchased from Sigma. Uncropped immunoblots are shown in Supplementary Fig. 8.

**Constructs**. All complementary DNAs were amplified by polymerase chain reaction (PCR) and cloned into either pDONR201 or pDONR221 vector according to the Gateway cloning procedure (Invitrogen). Entry clones were then recombined into Gateway-based destination vectors for the expression of N- or C-terminal-tagged fusion proteins. AUNIP deletion mutants were generated using the QuickChange Site-directed mutagenesis kit (Stratagene). All constructs used in this study were confirmed by DNA sequencing.

**Cell culture and transfection**. HeLa, U2OS, and HEK293T cells purchased from ATCC were cultured in Dulbecco's modified essential medium containing 10% fetal bovine serum and 1% penicillin and streptomycin and maintained at 37 °C in 5% $CO_2$. U2OS DR-GFP and U2OS EJ5-GFP cell lines were kindly provided by Dr. Maria Jasin (Memorial Sloan-Kettering Cancer Center) and Dr. Jeremy Stark (Beckman Research Institute of the City Hope), respectively. All these cell lines used in this study were confirmed to be free of mycoplasma contamination before use. For transient transfection experiments, cells were transfected with expression vectors using Lipofectamine 2000 (Invitrogen) according to the manufacturer's instructions. Small-interfering RNA (siRNA) duplexes targeting AUNIP (#1: CUUGUUU GCUAGACCGAAAUU; #2: CCAUUUGAUCCCAGGCUUAUU), CtIP (GCUAAAACAGGAACGAAUCdTdT), NBS1 (CCAACUAAAUUGCCAA-GUAU U), H2AX (CAACAAGAAGACGCGAAUCdTdT), RNF8 (ACUCAGU-GUCCAACUU GCUdTdT), and a nontargeting control siRNA (UUCAAUAAAUUCUUGAGGUUU) were purchased from Thermo Fisher Scientific. For siRNA-mediated depletion experiments, cells were transfected twice with siRNAs (100 nM) using Lipofectamine RNAiMAX (Invitrogen) according to the manufacturer's instructions.

**Tandem affinity purification (TAP)**. HEK293T cells were transfected with plasmids encoding SFB-tagged CtIP or AUNIP and were selected with medium containing 2 μg ml⁻¹ puromycin. Cell lines that stably express SFB-tagged CtIP or AUNIP were confirmed by immunoblotting and immunostaining experiments. To affinity purify CtIP- and AUNIP-protein complexes, engineered cells were lysed with NETN buffer (20 mM Tris-HCl [pH 8.0], 100 mM NaCl, 1 mM EDTA, and 0.5% Nonidet P-40) containing protease inhibitors (1 μg ml⁻¹ aprotinin, 1 μg ml⁻¹ leupeptin, and 100 mM PMSF) at 4 °C for 30 min. After centrifugation, the pellet was resuspended and sonicated in buffer (20 mM HEPES [pH 7.8], 0.4 M NaCl, 1 mM EDTA, 1 mM EGTA, and protease inhibitors) for 40 s to extract chromatin-bound proteins. The supernatants were then cleared by centrifugation at 15,000×g for 10 min at 4 °C to remove debris, and were subsequently incubated in the presence of with 150 μl of streptavidin-conjugated beads (GE Healthcare) for 2 h at 4 °C with gentle rocking. The immunocomplexes were washed three times with

NETN buffer and were eluted with 1 mg ml$^{-1}$ biotin (Sigma-Aldrich) for 1 h at 4 °C. The eluates were then incubated with 60 μl of S-protein Agarose beads (EMD Millipore) for 1 h at 4 °C. Proteins bound to S-protein Agarose were washed three times with NETN buffer, separated by sodium dodecyl sulfate polyacrylamide gel electrophoresis (SDS-PAGE), and analyzed by mass spectrometry.

**GST pull-down assays.** The coding sequence of wild-type CtIP was subcloned into pCold-MBP vector and transformed into BL21 E. Coli cells. At OD$_{600}$ 0.6, cells were induced with 0.1 mM IPTG at 16 °C for 16 h. Cells were then pelleted, resuspended in lysis buffer (20 mM Tris-HCl, 300 mM NaCl, 1% Triton X-100, and 1 μg ml$^{-1}$ each of leupeptin, aprotinin, and pepstatin) and sonicated on ice. The cell lysates were cleared by centrifugation at 40,000×g for 40 min at 4 °C and were incubated with Amylose resin for 2 h at 4 °C. After washing the beads with washing buffer (20 mM Tris-HCl, 500 mM NaCl, 0.5% NP-40, 1 mM DL-Dithiothreitol (DTT), and 1 μg ml$^{-1}$ each of leupeptin, aprotinin, and pepstatin), bound proteins were eluted with lysis buffer containing 10 mM Maltose. The coding sequences of wild-type and mutant AUNIP were subcloned into pCold-GST vector and transformed into E. Coli BL21 cells. At OD$_{600}$ 0.6, cells were induced with 0.2 mM IPTG at 16 °C for 16 h. Cells were then harvested, resuspended in lysis buffer (20 mM Tris-HCl, 300 mM NaCl, 1% Triton X-100, and 1 μg ml$^{-1}$ each of leupeptin, aprotinin, and pepstatin) and sonicated on ice. The cell lysates were cleared by centrifugation at 40,000×g for 40 min at 4 °C and incubated with glutathione-Sepharose resins for 2 h at 4 °C. After washing the beads with washing buffer (20 mM Tris-HCl, 500 mM NaCl, 0.5% NP-40, 1 mM DTT, and 1 μg ml$^{-1}$ each of leupeptin, aprotinin, and pepstatin), the bound proteins were used for pull-down assays. For in vitro pull-down assays, MBP-CtIP were incubated with GST or GST-AUNIP in NETN buffer for 2 h at 4 °C. The beads-bound proteins were then washed five times with NETN buffer and resolved on SDS-PAGE. MBP-CtIP was detected by a monoclonal anti-CtIP antibody.

**Co-immunoprecipitation and western blotting.** Cells were lysed with NETN buffer containing 20 mM NaF and protease inhibitors (1 μg ml$^{-1}$ aprotinin and leupeptin) on ice for 20 min. After centrifugation, the supernatants were incubated with either S-protein Agarose beads (EMD Millipore) or Protein A-Sepharose coupled with 2 μg of indicated antibodies for 4 h at 4 °C with gentle rocking. The beads-bound proteins were then washed three times with NETN buffer and were resolved on SDS-PAGE. Immunoblotting was performed according to standard procedures.

**HR and NHEJ reporter assays.** In all, $0.5 \times 10^6$ U2OS DR-GFP or U2OS EJ5-GFP cells were seeded in 6-well plates and were electroporated with 3 μg I-SceI expression plasmid (pCBASce) 24 h after. 48 h post pCBASce electroporation, cells were harvested and were subjected to flow cytometry analysis for GFP expression. Means were obtained from three independent experiments.

**CRISPR/Cas9 gene editing.** For CRISPR/Cas9-mediated knockout of AUNIP, the following guide RNAs (gRNAs) were used: AUNIP#1: GTGCTAAGCCTGG-GATCAAA and AUNIP#2: GGAGGCCTGCGGGCGTG TGGC. The gRNA sequences were cloned into the pX330-U6-Chimeric_BB-CBh-hSpCas9 vector (kindly provided by Dr. Feng Zhang) according to standard protocols[53, 54]. Cells were transfected with the gRNA/Cas9 expression construct and were selected in medium containing 2 μg ml$^{-1}$ puromycin. AUNIP expression was analyzed by western blotting with anti-AUNIP antibody.

**Immunofluorescence staining.** Cells were cultured on coverslips and were treated with 1 μM CPT or 10 Gy X-rays for the indicated times. Cells were then washed with PBS, permeabilized with PBS buffer containing 0.5% Triton X-100 for 5 min at room temperature, and were subsequently fixed with 3% paraformaldehyde for 10 min at room temperature. Coverslips were then blocked with 5% milk for 10 min before incubation with primary antibodies for 20 min at room temperature. Coverslips were washed three times with PBS before they were incubated with secondary antibodies for an additional 20 min at room temperature. Coverslips were then incubated with 4′,6-diamidino-2-phenylindole (DAPI) to visualize nuclear DNA. Images were captured with use of a fluorescence microscope (Eclipse 80i; Nikon) equipped with a Plan Fluor 60× oil objective lens (NA 0.5–1.25; Nikon) and a camera (CoolSNAP HQ2; PHOTOMETRICS).

**Detecting ssDNA lesions by BrdU incorporation.** Forty-eight hour post siRNA transfection, U2OS cells were incubated with 10 μM BrdU (Sigma) for 24 h, followed by treatment with 1 μM CPT for 1 h. After pre-extraction for 5 min in PBS containing 0.5% Triton X-100, cells were fixed with 3% paraformaldehyde solution for 10 min at room temperature. Subsequently, fixed cells were immunostained with anti-BrdU antibodies for 20 min at room temperature. Following three 5-min washes with PBS, cells were incubated with secondary antibodies for 20 min at room temperature. DAPI staining was then performed to visualize nuclear DNA.

**Laser micro-irradiation and live-cell imaging.** Laser micro-irradiation was performed as described previously[55]. Briefly, cells cultured on glass-bottomed 35-mm

dishes were micro-irradiated using a computer-controlled MicroPoint laser Ablation System (Photonics Instruments; 365 nm, 20 Hz) coupled to a Nikon Eclipse Ti-E inverted microscope (63× oil-immersion objective). Time-lapse images of live cells were taken under the same microscope with the MetaMorph Microscope Automation & Image Analysis software.

**Recombinant protein expression and purification.** The coding sequences of wild-type and mutant AUNIP were subcloned into pCold-MBP vector and transformed into BL21 E. Coli cells. At OD$_{600}$ 0.6, cells were induced with 0.1 mM IPTG at 16 °C for 16 h. Cells were then harvested, resuspended in lysis buffer (20 mM Hepes [PH 7.5], 300 mM NaCl, 1 mM DTT, and 1 μg ml$^{-1}$ each of leupeptin and aprotinin) and sonicated on ice. The cell lysates were cleared by centrifugation at 40,000×g for 40 min at 4 °C and incubated with Amylose resin for 4 h at 4 °C. After washing the beads with washing buffer (20 mM Hepes [PH 7.5], 500 mM NaCl, 1 mM DTT, and 1 μg ml$^{-1}$ each of leupeptin and aprotinin), bound proteins were eluted with lysis buffer containing 10 mM Maltose. Eluted proteins were dialyzed in dilution buffer (20 mM Hepes [PH 7.5], 1 mM DTT, and 1 μg ml$^{-1}$ each of leupeptin and aprotinin) and were loaded on pre-equilibrated 1 ml Hitrap Q HP column (GE Healthcare). Column was washed with 8 ml buffer A (20 mM Hepes [PH 7.5], 1 mM DTT and 1 μg ml$^{-1}$ each of leupeptin and aprotinin) and proteins were then eluted with a 50 ml gradient of buffer A containing 0–500 mM NaCl. Peak protein fractions were pooled and were concentrated with a 30-kDa Amicon Ultra centrifugal filter device (Millipore).

**Electrophoretic mobility shift assay (EMSA).** The EMSA assay was conducted using the Lightshift chemiluminescent EMSA kit (20148, Pierce) according to the manufacturer's instructions. Briefly, 20 nM biotin-labeled DNA substrates were incubated with indicated concentrations of AUNIP protein in 1× binding buffer containing 10 mM MgCl$_2$, 2.5% glycerol, and 0.05% Nonidet P-40 for 20 min at room temperature. Reactions were then stopped by the addition of 5 μl of gel loading buffer, resolved on a 10% native polyacrylamide gel, and transferred to a PVDF (polyvinylidene fluoride) membrane on ice. After UV (120 mJ per cm$^2$) cross-linking of the DNA to membrane, biotin-labeled DNA was detected using the Chemiluminescent Nucleic Acid Detection Module. The splayed arm substrate was generated by annealing oligo 1 (5′-GACGCTGCCGAATTCTACC AGTGCCTTG CTAGGACATCTTTGCCCACCTGCAG GTTCAC-3′) with oligo 2 (5′-ATAGT CGGATCCTCTAGACAGCTCCATGTAGCAAGGCACTGGTAGAATTCGG CAGCGTC-3′). The blunt-ended DNA substrate was generated by annealing oligo 1 with oligo 3 (5′-GTGAACCTGCAGGTGGGCAAAGATGTCCTAGCAAGGCA CTGGTAG AATTCG GCAGCGTC-3′). The 3′ tail DNA substrate was generated by annealing oligo 1 with oligo 4 (5′-TAGCAAGGCACTG GTAGAATTCGGCA GCGTC-3′). The 5′-flap structure was generated by annealing oligo 1 with oligo 2 and oligo 5 (5′-GTGAACCTG CAGGTGGGCAAAGATGTCC-3′). The 3′-flap structure was generated by annealing oligo 1 with oligo 2 and oligo 6 (5′-CATGG AGCTGTCTAGAGGATC CGACTAT-3′). Oligo 1 was 5′-biotin-labeled and the annealing reaction contains 250 nM oligo1 and 750 nM each of oligo 2, oligo 3, oligo 4, oligo 5, and oligo 6. All DNA substrates were purified by gel electrophoresis.

**Lentivirus packaging and infection.** AUNIP, CtIP, and NBS1 entry clones were transferred into a lentivirus-based, Gateway-compatible destination vector with N- or C-terminal GFP fusion tag. Lentiviruses were produced in HEK293T cells by co-transfection of the lentiviral-based construct with the packaging plasmids pMD2G and pSPAX2 (kindly provided by Dr. Songyang Zhou, Baylor College of Medicine). Forty-eight hour after transfection, infectious lentiviruses were harvested and were used for the transduction of HeLa or U2OS cells in the presence of 8 μg ml$^{-1}$ polybrene (Sigma). Stable cell pools were selected in medium containing 500 μg ml$^{-1}$ G418 (Calbiochem).

**Retrovirus production and infection.** Wild-type and mutant AUNIP entry clones were transferred into a Gateway-compatible retroviral destination vector pEF1A-HA-Flag. Retroviruses were produced in HEK293T cells by co-transfection of the retroviral plasmid with the packaging plasmids pCL-ECO and VSV-G. Forty-eight hour after transfection, viral supernatant were harvested and were used for the transduction of HeLa cells in the presence of 8 μg ml$^{-1}$ polybrene (Sigma). Stable pools were selected in medium containing 2 μg ml$^{-1}$ puromycin (Calbiochem).

**BrdU incorporation assays.** AUNIP knockout HeLa cells were split and transferred onto 60 mm dishes. Twenty-four hours later, cells were incubated with 100 μM BrdU for 1 h. Harvested cells were then washed with PBS, fixed in ice-cold 70% ethanol, and stored at 4 °C. DNA was subsequently denatured by using 2.5 M HCl for 1 h at room temperature. After three washes with PBS, cells were incubated with anti-BrdU antibody (Roche) diluted 1:100 in blocking buffer (PBS + 0.1% Triton X-100 + 5% BSA) for 12 h followed by incubation with the secondary FITC-conjugated anti-mouse antibody (1:100, Jackson immunoresearch) for 4 h at room temperature. Finally, cells were stained at 37 °C for 20 min with propidium iodide (20 μg ml$^{-1}$) and RNase A (200 μg ml$^{-1}$), and were analyzed on a FACScan flow cytometer (Beckman).

**Analysis of chromosomal aberrations**. HeLa cells were either mock treated or treated with 40 μM CPT for 12 h. Cells were then exposed to 1 μg ml$^{-1}$ colcemid for 4 h and were swollen using 75 mM KCl for 15 min at 37 °C. After fixing in methanol/acetic acid (3:1) (vol/vol) for 20 min, cells were dropped onto ice-cold wet slides, air dried, and were stained with 5% Giemsa for 5 min. The number of chromosome aberrations were scored in 50 metaphases per sample.

**Cell survival assays**. HeLa cells ($1 \times 10^3$) were seeded onto 60-mm dish in triplicates. At 24 h after plating, cells were treated with CPT, Topotecan, ABT-888, or IR at indicated doses. Twenty-four hours later, drug-containing medium was replaced with fresh medium and cells were cultured for additional 14 days at 37 °C to allow colony formation. Resulting colonies were then stained with Coomassie blue and counted.

**Data availability**. The data that support the findings of this study are available from the corresponding author upon request.

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

## Acknowledgements

We thank M. Jasin and J. Stark for U2OS DR-GFP and EJ5-GFP cell lines, and all our colleagues in the Huang laboratory for insightful discussions. This work was supported in part by the Key Program of the National Natural Science Foundation of China (31730021), National Natural Science Funds for Distinguished Young Scholar, National Program for Special Support of Eminent Professionals, National Basic Research Program of China Grant 2013CB911003, National Natural Science Foundation of China Grant 81661128008, 31571397 and 31071243, and the China's Fundamental Research Funds for the Central Universities.

## Author contributions

J.H., T.L., X.-H.F., M.S.Y.H., and J.L. designed the experiments; J.L., H.C., J.H.H., and H.H. performed the experiments; J.L., H.C., J.H.H., T.L., and J.H. analyzed the data; J.H. and T.L. wrote the manuscript.

## Additional information

**Competing interests:** The authors declare no competing financial interests.

