## [Peer Review File · Nature Communications]

Reviewers' comments:

Reviewer #1 (Remarks to the Author):

In this manuscript the authors identify a protein (c1orf135) that they rename SRAD, which they characterize as a regulator of DNA end resection and double-strand break repair. They specifically implicate SRAD in repair of breaks during DNA replication. They find that it localizes to DNA damage, binds CtIP, and regulates CtIP accumulation at breaks. They find mutants that interfere with its ability to bind to DNA or to bind CtIP. Both mutants are functionally defective. Overall I found the results to be convincing and largely support the author's model. The identification of SRAD and characterization of its function should be of interest to DNA repair investigators generally. The one major concern I have is that it is not clear that SRAD is really only a sensor of replication-associated breaks vs. a more general regulator of DSB repair. See specific comments below which should be addressed prior to publication.

1. The name "sensor of replication associated DSBs" implies that SRAD specifically functions at broken forks. The data for this specificity is not convincing. Most importantly, most of the localization studies are with laser induced damage which would not be fork-specific. The proteomics analysis of SRAD binding proteins did not pull out replisome proteins (other than perhaps DNA2 which could also be found at locations other than forks). The only data presented that it may be at forks is in figure 2a (and supplemental figure 2c) in which the authors claim SRAD is co-localized with RPA2 after CPT treatment. However, this data is not convincing. In fact, it looks like most of the SRAD foci are not co-localized with RPA2 foci. Either the authors need to more convincingly demonstrate that SRAD is functioning specifically at broken replication forks or they should revise the text (and perhaps the protein's name) to allow the possibility that SRAD more generally functions in controlling resection of DSBs irrespective of whether they happen at replication forks.

2. The statement that CtIP interacts with the SRAD C-terminus is an over-interpretation of the structure-function data. Deletion of the C-terminus indicates that it is required for the CtIP interaction but leaves open the possibility that the deletion alters the conformation of SRAD in such a way as to cause the loss of CtIP interaction. The authors would need to show that the C-terminus is sufficient to bind CtIP to know that it really contains the interaction surface.

Reviewer #2 (Remarks to the Author):

In this manuscript, Lou et al. describe a novel role for SRAD (previously called C1orf135/AIBP/AUNIP) in regulating DNA repair pathway choice. They show that SRAD interacts directly with CtIP, a key factor required for the initiation of homologous recombination (HR). Furthermore, SRAD localizes to DNA double-strand break (DSB) sites possibly by interacting with DNA directly, where it promotes the recruitment of CtIP to initiate DNA end resection. Consequently, cells lacking SRAD cannot produce the RPA-coated ssDNA template required for HR and are thus HR-deficient and sensitive to various types of genotoxic stress, particularly those that generate replication-associated DSBs.

DNA repair pathway choice is an area of intense research at the moment, so this work is highly topical and will be of general interest to a wide readership. I believe the authors' manuscript would be suitable for publication in Nature Communications with minor alterations (particularly to their model) and the addition of a few key experiments, as I detail below:

1. Although the authors have identified a novel role in DNA repair for SRAD, that does not mean they should rename a protein that has three names already. I accept the authors' argument that AIBP is

not appropriate as there is already a gene with that name. They should instead refer to it as AUNIP as that is the HUGO-approved name for it.

2. The authors show convincingly that the C-terminus of SRAD is required for binding to CtIP by pulldowns from cell extracts. They also show using purified proteins that the two proteins interact directly in vitro. However, to show that this is physiologically relevant they should also show whether recombinant SRAD 1-280 (missing the C-terminus) still interacts with CtIP in vitro. If the two proteins still bind then the in vitro interaction data should be removed from the manuscript as it would not represent the situation in vivo.

3. To complement the overexpression data, the authors should show the effect of SRAD loss on NHEJ using the EJ5-GFP system. If SRAD depletion or KO does not increase NHEJ levels, then the authors should remove their overexpression data with regard to HR and NHEJ as they could be looking at overexpression artifacts.

4. The authors claim that NBS1 recruitment in SRAD-deficient cells is not affected, and thus that SRAD is not affecting MRN recruitment to initiate resection. However, most of the cellular pool of MRN is associated with MDC1 and its recruitment is therefore H2AX/MDC1-dependent. The authors could therefore be missing a role for SRAD in promoting MRN recruitment to DNA ends to initiate resection. They therefore need to repeat the GFP-NBS1 recruitment experiments +/-SRAD in the absence of MDC1 or H2AX, otherwise they are not looking at the "resection-activating pool" of MRN.

5. The data with the DNA-PK inhibitor are not very convincing. S2056 is not necessarily a marker of increased NHEJ, it can also be elevated when NHEJ is blocked. Furthermore, DNA-PK inhibition causes defects in HR as well because it blocks Ku removal. The authors' rescue is marginal at best when using it. They should consider removing Figures 7h-i.

6. The authors need to reconsider their model to address the following points. Firstly, if SRAD and CtIP are constitutively interacting with each other, then how can it be that SRAD arrives at DSB sites before CtIP? Secondly, although SRAD does seem to have a preference for replication fork-like structures in vitro, it is clearly still binding to DSB ends as well. In addition, the authors show a very strong resection defect after X-rays, which induce resected DSB ends in S and G2 phase cells that are not at replication forks. It is therefore not clear that SRAD is specifically required for resection at replication-associated DSBs, so the authors need to modify their text and model accordingly. The title should also be modified to remove the phrase "replication-associated".

Other points:

1. P. 2, line 23 & p. 5, line 80: replace "the key determinant" & "the primary determinant" with "a key determinant" & "a primary determinant" respectively.

2. P. 3, line 43: NHEJ is not active throughout the cell cycle because it is inhibited during mitosis (Orthwein et al., Science 2014). Replace "throughout the cell cycle" with "throughout interphase".

3. P. 10, line 189: the authors mention SRAD silencing in reference to experiments in Figure 3, but in the figure panels are labelled "KO". The authors should clarify whether siRNAs or KO's are being used here.

4. It is difficult to make out the chromosomal breaks in Fig. 7A, and will be even more so when the figure is resized for publication. The authors should consider changing the image to make the data more accessible.

5. Supplemental figures S4 (h-j) are not mentioned anywhere in the main text.

6. The authors should provide the raw data for the mass spectrometry with tagged CtIP and SRAD as a supplemental file.

Point-by-point responses to the reviewers:

We would like to thank all the reviewers for their careful and constructive comments on our manuscript. We have now conducted a number of experiments as suggested by the reviewers. We hope that these new experimental findings address reviewers' concerns and provide additional support to our main conclusions. According to the reviewer's suggestion, we have also revised the text and model. Below is our point-by-point response to the reviewers' critiques:

Reviewer #1 (Remarks to the Author):

In this manuscript the authors identify a protein (c1orf135) that they rename SRAD, which they characterize as a regulator of DNA end resection and double-strand break repair. They specifically implicate SRAD in repair of breaks during DNA replication. They find that it localizes to DNA damage, binds CtIP, and regulates CtIP accumulation at breaks. They find mutants that interfere with its ability to bind to DNA or to bind CtIP. Both mutants are functionally defective. Overall I found the results to be convincing and largely support the author's model. The identification of SRAD and characterization of its function should be of interest to DNA repair investigators generally.

Thanks for the nice summary!

The one major concern I have is that it is not clear that SRAD is really only a sensor of replication-associated breaks vs. a more general regulator of DSB repair. See specific comments below which should be addressed prior to publication.

1. The name "sensor of replication associated DSBs" implies that SRAD specifically functions at broken forks. The data for this specificity is not convincing. Most importantly, most of the localization studies are with laser induced damage which would not be fork-specific. The proteomics analysis of SRAD binding proteins did not pull out replisome proteins (other than perhaps DNA2 which could also be found at locations other than forks). The only data presented that it may be at forks is in figure 2a (and supplemental figure 2c) in which the authors claim SRAD is co-localized with RPA2 after CPT treatment. However, this data is not convincing. In fact, it looks like most of the SRAD foci are not co-localized with RPA2 foci. Either the authors need to more convincingly demonstrate that SRAD is functioning specifically at broken replication forks or they should revise the text (and perhaps the protein's name) to allow the possibility that SRAD more generally functions in controlling resection of DSBs irrespective of whether they happen at replication forks.

Thanks for your suggestion. The same issue was also raised by Reviewer #2. We have now revised the text and model as suggested. In addition, we have removed the phrase “replication-associated” from the Manuscript Title. Finally, we used the HUGO-approved name “AUNIP” in the revised manuscript to avoid any confusion in the field.

2. The statement that CtIP interacts with the SRAD C-terminus is an over-interpretation of the structure-function data. Deletion of the C-terminus indicates that it is required for the CtIP interaction but leaves open the possibility that the deletion alters the conformation of SRAD in such a way as to cause the loss of CtIP interaction. The authors would need to show that the C-terminus is sufficient to bind CtIP to know that it really contains the interaction surface.

As suggested, we have now performed *in vitro* GST pull-down experiments using purified recombinant proteins and showed that the C-terminus of AUNIP is sufficient to bind CtIP (Please see the revised Figure 1g).

Reviewer #2 (Remarks to the Author):

In this manuscript, Lou et al. describe a novel role for SRAD (previously called C1orf135/AIBP/AUNIP) in regulating DNA repair pathway choice. They show that SRAD interacts directly with CtIP, a key factor required for the initiation of homologous recombination (HR). Furthermore, SRAD localizes to DNA double-strand break (DSB) sites possibly by interacting with DNA directly, where it promotes the recruitment of CtIP to initiate DNA end resection. Consequently, cells lacking SRAD cannot produce the RPA-coated ssDNA template required for HR and are thus HR-deficient and sensitive to various types of genotoxic stress, particularly those that generate replication-associated DSBs.

DNA repair pathway choice is an area of intense research at the moment, so this work is highly topical and will be of general interest to a wide readership. I believe the authors' manuscript would be suitable for publication in Nature Communications with minor alterations (particularly to their model) and the addition of a few key experiments, as I detail below:

Thanks for the nice summary!

1. Although the authors have identified a novel role in DNA repair for SRAD, that does not mean they should rename a protein that has three names already. I accept the authors' argument that AIBP is not appropriate as there is already a gene with that name. They should instead refer to it as AUNIP as that is the HUGO-approved name for it.

Agree. We have now used the HUGO-approved name “AUNIP” in the revised manuscript to avoid any confusion in the field.

2. The authors show convincingly that the C-terminus of SRAD is required for binding to CtIP by pulldowns from cell extracts. They also show using purified proteins that the two proteins interact directly *in vitro*. However, to show that this is physiologically relevant they should also show whether recombinant SRAD 1-280 (missing the C-terminus) still interacts with CtIP *in vitro*. If the two proteins still bind then the *in vitro* interaction data should be removed from the manuscript as it would not represent the situation *in vivo*.

Thanks for your suggestion! We have now performed *in vitro* GST pull-down assays using purified recombinant proteins and found that recombinant AUNIP 1-280 does not interact with CtIP *in vitro* (Please see revised Figure 1g). In addition, we demonstrated that the C-terminus of AUNIP is sufficient to bind CtIP (Please see revised Figure 1g). Together, our results suggest that the C-terminus of AUNIP is necessary and sufficient for binding to CtIP.

3. To complement the overexpression data, the authors should show the effect of SRAD loss on NHEJ using the EJ5-GFP system. If SRAD depletion or KO does not increase NHEJ levels, then the authors should remove their overexpression data with regard to HR and NHEJ as they could be looking at overexpression artifacts.

As suggested by the reviewer, we examined the effect of AUNIP silencing on NHEJ using the EJ5-GFP system. As shown in the revised Supplementary Figure 3b-c, downregulation of AUNIP led to a dramatic increase in the frequency of NHEJ. These results are consistent with our original hypothesis in which AUNIP promotes HR and inhibits NHEJ.

4. The authors claim that NBS1 recruitment in SRAD-deficient cells is not affected, and thus that SRAD is not affecting MRN recruitment to initiate resection. However, most of the cellular pool of MRN is associated with MDC1 and its recruitment is therefore H2AX/MDC1-dependent. The authors could therefore be missing a role for SRAD in promoting MRN recruitment to DNA ends to initiate resection. They therefore need to repeat the GFP-NBS1 recruitment experiments +/-SRAD in the absence of MDC1 or H2AX, otherwise they are not looking at the “resection-activating pool” of MRN.

Thanks for your suggestion! We have now performed additional microirradiation experiments to explore the role of AUNIP in the recruitment of NBS1 to sites of DNA damage in the presence or absence of H2AX. As shown in the revised Supplementary

Figure 5, loss of AUNIP has no significant effect on the recruitment of NBS1 to sites of laser-induced DNA damage in either the presence and absence of H2AX.

5. The data with the DNA-PK inhibitor are not very convincing. S2056 is not necessarily a marker of increased NHEJ, it can also be elevated when NHEJ is blocked. Furthermore, DNA-PK inhibition causes defects in HR as well because it blocks Ku removal. The authors' rescue is marginal at best when using it. They should consider removing Figures 7h-i.

We agree with the Reviewer and have now removed Figures 7h-i from the revised manuscript.

6. The authors need to reconsider their model to address the following points. Firstly, if SRAD and CtIP are constitutively interacting with each other, then how can it be that SRAD arrives at DSB sites before CtIP? Secondly, although SRAD does seem to have a preference for replication fork-like structures in vitro, it is clearly still binding to DSB ends as well. In addition, the authors show a very strong resection defect after X-rays, which induce resected DSB ends in S and G2 phase cells that are not at replication forks. It is therefore not clear that SRAD is specifically required for resection at replication-associated DSBs, so the authors need to modify their text and model accordingly. The title should also be modified to remove the phrase "replication-associated".

Thanks for your suggestion. The same issue was also raised by Reviewer #1. We have now revised the text and model as suggested. In addition, we have removed the phrase "replication-associated" from the Manuscript Title as suggested by the Reviewer.

Other points:

1. P. 2, line 23 & p. 5, line 80: replace "the key determinant" & "the primary determinant" with "a key determinant" & "a primary determinant" respectively.

Thanks. These have now been corrected in the revised manuscript.

2. P. 3, line 43: NHEJ is not active throughout the cell cycle because it is inhibited during mitosis (Orthwein et al., Science 2014). Replace "throughout the cell cycle" with "throughout interphase".

Thanks for the clarification and we have now modified the text accordingly and cited this paper in the revised manuscript.

3. P. 10, line 189: the authors mention SRAD silencing in reference to experiments in Figure 3, but in the figure panels are labelled “KO”. The authors should clarify whether siRNAs or KO’s are being used here.

We apologize for the lack of clarity. AUNIP knockout DR-GFP U2OS cells are used here. We have now amended the text and Figure Legend accordingly.

4. It is difficult to make out the chromosomal breaks in Fig. 7A, and will be even more so when the figure is resized for publication. The authors should consider changing the image to make the data more accessible.

As suggested by the Reviewer, we have now replaced these images in our revised Figure 7a.

5. Supplemental figures S4 (h-j) are not mentioned anywhere in the main text.

Sorry, this mistake has now been corrected in the revised manuscript.

6. The authors should provide the raw data for the mass spectrometry with tagged CtIP and SRAD as a supplemental file.

As suggested by the Reviewer, we have now provided the raw data in the revised version of our manuscript (Please see revised Supplementary Data 1 and Data 2).

Reviewers' comments:

Reviewer #1 (Remarks to the Author):

The one remaining concern is that the conclusion that AUNIP is localized with RPA to damaged replication forks is not adequately supported. As I mentioned in my original review, the immunofluorescence images showing co-localization in foci (now supplemental figure 2c) are not convincing. At a minimum, the authors need to provide a merged green/red image and some quantitation of the co-localization. I also think they should change the wording in the manuscript (line 154) since the data presented do not show that the RPA and AUNIP foci "coincide".

Reviewer #2 (Remarks to the Author):

The authors have satisfactorily addressed all the points in my previous review and I wish to congratulate them on a very interesting story.

Minor point: please include details of the GST-pulldown method as it is missing from the experimental procedures.

Reviewer #1 (Remarks to the Author):

The one remaining concern is that the conclusion that AUNIP is localized with RPA to damaged replication forks is not adequately supported. As I mentioned in my original review, the immunofluorescence images showing co-localization in foci (now supplemental figure 2c) are not convincing. At a minimum, the authors need to provide a merged green/red image and some quantitation of the co-localization. I also think they should change the wording in the manuscript (line 154) since the data presented do not show that the RPA and AUNIP foci "coincide".

Thanks for your suggestion. We have now provided merged images (Please see Figure 2a and Supplementary Figure 2d) and quantitation of the co-localization in the revised manuscript (Please see Supplementary Figure 2c). The Reviewer is correct in pointing out that RPA and AUNIP foci do not overlap entirely. Interestingly, AUNIP foci are exclusively found in cells that also supported formation of RPA2 foci. We speculate that this may reflect that RPA binds ssDNA and is able to spread to adjacent areas of damaged chromatin. By contrast, AUNIP preferentially binds to DNA substrates that mimic structures generated at stalled replication forks, and thus may only accumulate proximal to stalled replication forks. In addition, as suggested by this Reviewer, we have changed the wording in the revised manuscript.

Reviewer #2 (Remarks to the Author):

The authors have satisfactorily addressed all the points in my previous review and I wish to congratulate them on a very interesting story.

Thanks!

Minor point: please include details of the GST-pulldown method as it is missing from the experimental procedures.

Thanks for your suggestion. We have now included details of the GST pull-down method in the revised manuscript.